# Recent Developments in Targeting the Cell Cycle in Melanoma

**DOI:** 10.3390/cancers17081291

**Published:** 2025-04-11

**Authors:** Christie Hung, Trang T. T. Nguyen, Poulikos I. Poulikakos, David Polsky

**Affiliations:** 1Ronald O. Perelman Department of Dermatology, New York University Grossman School of Medicine, Laura and Isaac Perlmutter Cancer Center, NYU Langone Health, New York, NY 10016, USA; christie.hung@nyulangone.org (C.H.); thithutrang.nguyen@nyulangone.org (T.T.T.N.); 2Department of Oncological Sciences, The Tisch Cancer Institute, Icahn School of Medicine at Mount Sinai, New York, NY 10029, USA; poulikos.poulikakos@mssm.edu

**Keywords:** melanoma, cell cycle, cyclin D, CDK2, CDK4, CDK6, RB, PI3k, Akt, mTOR, inhibitor, immunotherapy

## Abstract

Melanoma is an aggressive and increasingly common cancer. Despite advances in targeted therapies and immunotherapies, many patients fail to respond or develop resistance, highlighting the need for new treatments. The Cyclin D–CDK4/6–RB1 pathway, altered in up to 90% of melanoma cases, offers a potential target. However, therapies targeting Cyclin–CDK complexes have shown limited success. Our review examines the role of Cyclin–CDK circuitry in cell cycle regulation and explores clinical trials focusing on combination therapies using CDK4/6 inhibitors. This approach aims to improve outcomes for molecularly defined subsets of melanoma patients.

## 1. Introduction

While there have been advancements in treating metastatic melanoma using BRAF-MEK targeted therapies and immunotherapies, effective treatments remain elusive for patients for whom these therapies have failed [1]. Specifically, there is an urgent demand for new therapies aimed at patients with BRAF wild-type tumors who advance despite immunotherapy and those with BRAF-mutant melanoma who progress on both immunotherapy and BRAF-MEK targeted treatments. Cyclin-dependent kinases (CDKs) and their regulatory partners, Cyclins, are fundamental drivers of cell cycle progression. Specifically, CDK4 and CDK6 regulate the G1-to-S phase transition by forming complexes with D-type Cyclins (Cyclin D1, D2, or D3) to phosphorylate and inactivate the retinoblastoma (RB) protein. This phosphorylation releases E2F transcription factors, promoting the expression of genes required for DNA replication and cell cycle progression. The dysregulation of the CDK4/6 pathway is commonly observed in various cancers, leading to uncontrolled cell proliferation, making CDK4/6 a key therapeutic target in oncology [2]. Small-molecule inhibitors simultaneously targeting CDK4 and CDK6 (collectively, CDK4/6) are established as standard treatments when used in combination with endocrine therapy for hormone receptor (HR)-positive, human epidermal growth factor receptor 2 (HER2)-negative breast cancer [3]. Unfortunately, CDK4/6 inhibitors (CDK4/6is) have demonstrated limited clinical effectiveness in melanoma to date. This is similar to the situation in other cancers, such as lung and colorectal cancers, and glioblastoma [4,5,6]. Given that the dysregulation of the key mitogenic pathways that regulate Cyclin D is a hallmark of melanoma, we would expect therapies targeting CDK4/6 to have greater clinical efficacy than has been observed [7]. In the last several years, preclinical studies have revealed a greater complexity of the cell cycle regulatory circuit than was originally described, illuminating potential new avenues to block melanoma cell proliferation. Here, we provide an update on developments that may lead to more effective CDK4/6i combination therapies targeting the cell cycle for the second- or third-line treatment of melanoma.

## 2. The Role of the Cyclin–CDK-RB Axis in Regulating Cell Proliferation

In 2001, Leland H. Hartwell, R. Timothy Hunt, and Paul M. Nurse were awarded the Nobel Prize for their work elucidating the roles of Cyclins and CDKs, pivotal regulatory molecules governing the cell cycle. In the early 1970s, Leland H. Hartwell identified key cell cycle regulatory genes in yeast, including the “start” gene (CDC28), which plays a crucial role in initiating cell division [8,9]. In the early 1980s, Paul M. Nurse discovered CDK1, one of several human homologs to yeast CDC28, demonstrating the conservation of the cell cycle regulatory machinery across species [10]. In the mid-1980s, R. Timothy Hunt, working on sea urchins, discovered Cyclins, the proteins that regulate CDKs to ensure the proper timing of cell cycle progression [11]. In the early 1990s, Charles J. Sherr and other investigators identified mammalian D-type G1 Cyclins, followed by the CDKs with which they were associated: CDK4 and CDK6 [12,13,14,15]. Mitogenic signaling through the RAS-ERK and PI3K-mTOR networks leads to the upregulation of D-type Cyclins (Cyclins D1, D2, and D3), primarily through increased transcription and decreased proteasomal degradation (Figure 1) [16,17,18]. Cyclin D1 binds to and activates CDK4; Cyclin D3 binds to and activates CDK6 [19,20].

A major function of Cyclin D–CDK4/6 complexes is to phosphorylate the retinoblastoma (RB) tumor suppressor protein (Figure 2). In its hypophosphorylated form, RB acts as a transcriptional repressor by binding to E2F transcription factors and inhibiting E2F target gene expression and the G1/S cell cycle transition [21]. Phosphorylated RB dissociates from E2F, allowing for the transcription of E2F-responsive genes, including Cyclins E1 (CCNE1) and E2 (CCNE2). Both Cyclins E1 and E2 bind to and activate CDK2. Cyclin E–CDK2 complexes hyperphosphorylate RB, which drives the expression of additional genes including Polo-like kinase 1 (PLK1, involved in mitosis), Cyclin A (critical for DNA replication initiation) when bound to CDK2, dihydrofolate reductase (DHFR), thymidylate synthase (TYMS), and DNA polymerase alpha. These genes collectively play critical roles in mediating an irreversible cellular commitment to the S phase, ensuring proper DNA replication [22]. The stage within the G1 phase where the cell becomes irreversibly committed to the S phase is known as the “restriction point”, which is considered an important checkpoint in the cell when cells are independent of further mitogenic stimuli for entry into the S phase [23]. Upon the completion of mitosis, the classical view has been that RB returns to its dephosphorylated state and binds to E2F, inhibiting the expression of genes driving cell cycle progression after mitosis. However, more recent studies support the concept that under the appropriate conditions, cells can maintain phosphorylated RB and continuously proliferate [24].

Another level of regulation involves endogenous CDK inhibitor proteins, which are classified into two families: CIP/KIP and INK4 (Figure 2) [18]. The CIP/KIP proteins p21CIP1, p27KIP1, and p57KIP2 are encoded by the CDKN1A, B, and C genes, respectively, and primarily inhibit the kinase activity of the Cyclin E–CDK2 (KIP1, CIP1) and Cyclin A–CDK2 (KIP2) complexes. The INK4 proteins p16INK4A, p15INK4B, p18INK4C, and p19INK4D are encoded by the CDKN2A, B, C, and D genes, respectively, and individually bind to monomeric CDK4 and CDK6 proteins.

## 3. Alterations to the Cyclin–CDK-RB Axis in Melanoma

A hallmark of melanoma is the nearly universal dysregulation of cell cycle control pathways resulting from genetic or gene expression alterations, demonstrating the critical role of this axis in melanoma cell proliferation. For example, approximately 40% of melanoma-prone families possess inactivating germline mutations in CDKN2A, which encodes p16INK4A [25,26] (Figure 1). Germline mutations in CDKN2A have also been associated with an increased risk for pancreatic cancer which can co-occur in a subset of melanoma-prone families [25]. The germline CDK4 R24C mutation is also found in melanoma-prone families, albeit rarely. The R24C mutation prevents p16INK4A from binding to CDK4, allowing for uninhibited cell cycle progression through RB phosphorylation by Cyclin D–CDK4 complexes [27]. Somatically, CDKN2A is frequently inactivated by deletion, methylation, or mutation [28], while CDK4 mutations, RB1 deletions, and CCNE1 copy number gains are rare. Cyclin D1 (CCND1) and CDK4 copy number gains often appear in acral melanoma, suggesting a pathogenic role [29].

The activation of the MAPK and PI3K pathways is nearly universal in melanoma tumors principally due to mutually exclusive mutations in BRAF (40–50% of cases) or NRAS (15–20%) or to the losses of NF1 expression (13%) that may co-occur with BRAF or NRAS mutations [30,31]. NF1, a tumor suppressor and GTPase-activating protein (GAP), negatively regulates RAS signaling by converting RAS-GTP to its inactive form, RAS-GDP. However, when NF1 is inactivated, the continuous and unchecked activation of RAS amplifies downstream signaling cascades, particularly through the MAPK and PI3K-AKT pathways, which are well known for their roles in promoting cell proliferation and survival [32]. As a result of these upstream alterations, the hyperactivation of ERK is observed in approximately 90% of melanomas [33]. Signaling through ERK leads to the upregulation of transcription factors such as Fos, Jun, and ATF, which in turn drive CCND1 expression [34]. Additionally, Cyclin D1 translation is enhanced by the activation of the PI3K-Akt pathway (Figure 2) [33,35,36]. This dual pathway activation underscores Ras’s role in supporting melanoma cell cycle advancement and tumor growth.

## 4. Preclinical and Clinical Studies in Melanoma Utilizing CDK4/6 Inhibitors

Three CDK4/6 inhibitors, palbociclib, ribociclib, and abemaciclib, have been approved for use in the US for patients with hormone receptor-positive (HR+), human epidermal growth factor receptor 2-negative (HER2-) breast cancer in combination with endocrine therapy [37]. These inhibitors were explored in melanoma based on compelling preclinical evidence demonstrating their critical role in cell cycle regulation. The CDK4/6 pathway is frequently dysregulated in melanoma, particularly in tumors with intact retinoblastoma (RB) protein, which comprise the large majority of melanomas [29]. Many preclinical studies have shown that CDK4/6 inhibitors can effectively induce cell cycle arrest in melanoma cells, providing a strong foundation for their clinical evaluation. Additionally, acral melanomas frequently harbor copy number gains in CCND1 and CDK4 and losses of CDKN2A, and a large preclinical study showed that palbociclib monotherapy was effective in treating acral melanoma patient derived xenograft tumors with various alterations to Cyclin–CDK circuitry [38]. Other preclinical studies explored using CDK4/6 inhibitors as monotherapy or in combination with MEK inhibitors in the setting of acquired resistance to BRAF inhibitors. One study identified acquired oncogenic mutations in NRAS and the loss of CDKN2A in BRAF-V600E-mutant melanoma tumors from two patients who progressed on BRAF/MEK inhibitors, suggesting that the reactivation of MAPK and the activation of cell cycle pathways contribute to resistance. Combining CDK4/6 (palbociclib) and a MEK inhibitor (trametinib) overcame this resistance, offering a promising strategy for patients with metastatic melanoma who are refractory to BRAF/MEK therapy [39]. Another study found that palbociclib could induce senescence in vemurafenib-resistant melanoma cell lines in vitro and when grown in immunocompromised mice [40].

Despite these encouraging preclinical studies, early clinical studies in melanoma evaluated CDK4/6is as monotherapy and demonstrated limited efficacy (Table 1) [41]. A phase 1 clinical trial (NCT01394016) of abemaciclib monotherapy that included 26 patients with advanced metastatic melanoma showed a disease control rate of only 27% [42]. A phase 2 trial of palbociclib (NCT03454919) in 15 patients with advanced acral melanoma and CDK4 pathway alterations found that 20% achieved some tumor shrinkage at 8 weeks, but only 1 was of sufficient magnitude to be considered a partial response [43]. Although disappointing, these results are unsurprising because CDK4/6is are not used as monotherapy in other cancers, including breast cancer where it is most often used in combination with hormone therapy. In fact, recent results from the NCI-MATCH trial, which included a sub-study to assess palbociclib in patients with solid tumors or lymphomas harboring at least seven copies of CDK4 or CDK6, found that only 1 of 38 patients had a partial response, 10 had stable disease, and 21 experienced disease progression, with responses occurring only in CDK4-amplified (but not CDK6-amplified) cases. The median progression-free survival was 2.0 months, and overall survival was 8.8 months, indicating limited efficacy [44]. Another consideration when interpreting these results is that all of these trials were conducted in patients with advanced-stage disease following the failure of first-line and possibly subsequent treatments. Patients in this setting typically have very aggressive tumors that have developed resistance mechanisms, possibly including additional acquired mutations or chromosomal abnormalities, that may have contributed to poor clinical responses. These findings emphasize the need to consider both the timing of treatment and the molecular context of the disease when evaluating clinical trial outcomes.

Given that the MAPK pathway is frequently activated in melanoma through mutations in BRAF, combining CDK4/6is with BRAF inhibitors (BRAFis) and MEK inhibitors (MEKis) offered a promising therapeutic strategy. A phase 1b/2 dose escalation trial in metastatic melanoma (NCT01543698) combined ribociclib (CDK4/6i) with encorafenib (BRAFi) and binimetinib (MEKi) [45]. Unfortunately, the triple combination did not prolong progression-free survival (PFS) compared to dual encorafenib (BRAFi) and binimetinib (MEKi) treatment [45]. Another phase 1b/2 trial (NCT01781572) of 41 patients examined the safety and efficacy of ribociclib in combination with binimetinib with NRAS-mutant melanoma and investigated biomarkers’ associations with response [46]. The authors observed an overall response rate for the combination of 19.5%; however, the response rate was 32.5% for patients with alterations in CDKN2A, CDK4, or CCND1. Notably, these response rates were achieved using a much lower dose of ribociclib than that used in breast cancer due to the toxicity of the combination in the phase 1b part of the study. Another phase 1/2 trial (NCT02202200) investigated palbociclib (CDK4/6i) alongside vemurafenib (BRAFi) in 18 BRAFV600-mutant metastatic melanoma patients who received previous BRAFi treatment. The combination demonstrated a clinical response rate of 27.8% [47]. These results suggest that adding CDK4/6is to BRAFis may provide therapeutic benefits for patients who develop resistance to BRAFis alone. Similar to the prior study that combined ribociclib with binimetinib, toxicity from the combination therapy was a major concern. The maximum tolerated dose of palbociclib was only 25 mg daily for two weeks followed by one week off compared to 125 mg daily for three weeks followed by one week off in breast cancer. Surprisingly, the major toxicity was severe vemurafenib-associated skin rash and not neutropenia typically associated with palbociclib [47].

Other trials have also evaluated co-targeting the MAPK pathway with CDK4/6 inhibitors. A phase 1b clinical trial of trametininb (MEKi) and palbociclib (CDK4/6i) included one patient with advanced melanoma who experienced a partial response of at least 13.6 months; however, substantial toxicities limited the further development of this combination therapy (NCT02065063) [48]. A phase 1 trial of an ERK1/2 inhibitor with other agents including abemaciclib for advanced or metastatic solid tumors was open to melanoma patients; however, the efficacy results have not yet been published (NCT02857270). The LOGIC-2 trial (NCT02159066) evaluated the efficacy of adding one of four other agents including ribociclib (CDK4/6i) to encorafenib (BRAFi) and binimetinib (MEKi) in 38 patients with advanced BRAF V600-mutated melanoma who progressed on encorafenib and binimetinib. They found that the objective response rate was 5.3% with a median progression-free survival of 2.1 months. No new safety signals were observed [49]. Taken together, these studies demonstrate the challenges of combining CDK4/6is with BRAFi and/or MEKi therapy, particularly toxicity, which often necessitates dose reductions or even treatment discontinuation. This limitation underscores the difficulty in achieving an optimal therapeutic dose in combination regimens, further complicating their clinical utility. Consequently, there is an urgent need to identify novel drug combinations that not only maintain therapeutic efficacy but also minimize toxicity, ultimately improving patient outcomes and expanding viable treatment options.

## 5. Redundancies in CDK Pathways: Implications for Cancer Therapy Resistance

Given the clinical trial results above and recent discoveries highlighting functional redundancies among CDKs, there may be other proteins that can be targeted in combination with CDK4/6is to effectively arrest the cell cycle. Studies have demonstrated that CDK4/6 is not required for RB phosphorylation prior to S phase entry. Knockout cancer cell lines and murine models lacking CDK2 or CDK4/6 expression revealed that each of these kinases can independently drive cellular proliferation [50,51]. This redundancy of Cyclin–CDK complexes controlling cell cycle progression has been identified as an important mechanism of resistance to CDK4/6i therapies [7,41]. In other words, in the presence of CDK4/6 inhibition, CDK2 can compensate for CDK4/6.

Adding to this complexity, studies revealed that the members of the CIP/KIP family, such as p21Cip1, p27Kip1, and p57Kip2, both inhibit and promote the assembly of Cyclin D–CDK4/6 complexes. At low levels, they aid in complex assembly, supporting cell cycle progression from the G1 to S phase via RB phosphorylation. However, at higher levels, they inhibit these complexes, causing cell cycle arrest. This dual function helps regulate controlled cell division and prevent excessive proliferation, a process disrupted in cancer (Figure 2) [18]. The amplification or overexpression of CDK4/6 or Cyclin D can lead to the sequestration of p21Cip1 and p27Kip1, preventing them from inhibiting Cyclin E–CDK2 complexes, contributing to aberrant cell proliferation [52]. CDK2 inhibitors or CDK2/4/6 inhibitors currently in development may be one therapeutic approach to overcoming this mechanism of resistance [see Section 7].

Another function of Cyclin D–CDK4/6 complexes is to phosphorylate methylosome protein 50 (MEP50), a critical coactivator of protein arginine methyltransferase 5 (PRMT5) (Figure 2). Once activated, PRMT5 methylates proteins vital to the spliceosome machinery, ensuring the accurate splicing of various transcripts, including mouse double minute 4 (MDM4), and increasing their expression [53]. Elevated MDM4 expression inhibits TP53 (p53) activity, reducing the levels of p21Cip1, a suppressor of the CDK2–Cyclin E complex. This leads to enhanced CDK2 activity, increased RB1 phosphorylation, and the subsequent activation of E2F-mediated gene transcription (Figure 2) [53,54]. Altogether, these evolving findings describing Cyclin–CDK circuitry reveal its complexity and provide insights that may guide the development of future treatments targeting the cell cycle.

## 6. Relative Levels of CDK4 and CDK6 Impact Efficacy of CDK4/6is in Melanoma

Some recent studies have suggested that CDK4/6i effectiveness depends on the CDK4-to-CDK6 ratio. Pack et al. showed that CDK4/6 inhibitors like palbociclib have both catalytic and non-catalytic roles—blocking RB phosphorylation and displacing p21CIP1 from CDK4, allowing it to inhibit CDK2. However, palbociclib failed to dissociate p21CIP1 from CDK6 complexes, implying that high CDK6 levels may trap p21CIP1, preventing it from inhibiting CDK2, ultimately reducing the effectiveness of CDK4/6 inhibitors [55].

A study by Wu et al. revealed that cellular sensitivity to palbociclib depends on the conformation of CDK6. The knockdown of CDK6 sensitized melanoma cells to palbociclib, and binding was observed only when CDK6 was in its normal conformation, associated with the CDC37-HSP90 chaperone complex. In resistant cells, CDK6 adopted a different conformation, preventing palbociclib binding. No CDK6 mutations were identified, leaving the determinants of these conformational changes unknown [56,57]. Together, these papers suggest that strategies to decrease CDK6 levels in melanoma may improve response to CDK4/6is. One strategy could involve the use of CDK6 proteolysis targeting chimeras (PROTACs) that selectively degrade their target proteins [56,58,59,60,61,62].

Contrary to previous studies that identified high CDK6 levels as a potential determinant of resistance to CDK4/6is, Wang et al. found that high CDK6 levels can increase sensitivity to CDK4/6 inhibitors through an RB-independent metabolic mechanism. In T-cell acute lymphoblastic leukemia (T-ALL) tumors, they demonstrated that high levels of Cyclin D3–CDK6 were associated with an increased expression of antioxidant proteins such as glutathione and NADPH that protect cells against oxidative stress. In melanoma patient-derived xenografts, they found that 3 out of 33 melanoma tumors treated with ribociclib monotherapy underwent regression instead of growth arrest. These three cases had high Cyclin D3–CDK6 levels (in contrast to the levels detected in non-regressing tumors), and ribociclib treatment depleted NADPH and glutathione, resulting in increased ROS [63]. These data suggest that Cyclin D3–CDK6 supports survival via metabolism, and its inhibition can lead to tumor shrinkage, highlighting the impact of CDK4/6 balance on CDK4/6 inhibitor efficacy.

## 7. Combining CDK2 and CDK4/6 Inhibitors for Synergistic Anti-Cancer Therapy

A more recent approach to improving the antitumor efficacy of CDK4/6is in cancer focused on the functional redundancies in the Cyclin–CDK circuitry. One study examining CDK dependencies in RB-proficient cancer cell types identified a subset of cell lines that relied on Cyclin E–CDK2 complexes for RB phosphorylation, independent of CDK4/6 activity, as evidenced by their resistance to cell cycle arrest from targeted deletions of CDK4 and CDK6 [64]. This redundancy in the control of RB phosphorylation suggests that the effectiveness of CDK4/6is may be improved in some cancers by combining CDK2 inhibition with CDK4/6is [7,65]. Building on this, Freeman-Cook and colleagues developed PF3600, a pan-CDK2/4/6 inhibitor, to address key resistance mechanisms to CDK4/6 inhibitors [66]. For example, Cyclin E amplification is commonly observed to be a driver of therapeutic resistance in hormone receptor-positive (HR+)/HER2-negative breast cancers. Preclinical evaluations using in vivo xenograft models demonstrated that PF3600 effectively targets the Cyclin E–CDK2 complex, showing promise in overcoming resistance to established CDK4/6 inhibitors like palbociclib [66,67,68]. Additionally, studies on gastrointestinal stromal tumor (GIST) cell lines with intact RB revealed that the dual inhibition of CDK2 (PF-06873600, CDK2 inhibitor II) and CDK4/6 (palbociclib or abemaciclib) effectively arrested cells in the G1 phase [67]. Despite its promising in vitro efficacy, PF-06873600’s clinical development was discontinued due to significant toxicity, underscoring the challenges in simultaneously targeting multiple Cyclin-dependent kinases (CDK2/4/6) to treat human cancers. Newer strategies are focusing on combining selective CDK2 inhibitors with CDK4/6 inhibitors to optimize dosing, reduce toxicity, and enhance therapeutic efficacy. Notable CDK2 inhibitors under clinical investigation include PF-07104091 (Pfizer (New York, NY, USA), NCT04553133), BLU-222 (Blueprint Medicines (Cambridge, MA, USA), NCT05252416), INX-315 (Incyclix Bio (Durham, NC, USA), NCT05735080), AZD8421 (AstraZeneca (Wilmington, DE, USA), NCT06188520), and INCB123667 (Incyte (Wilmington, DE, USA), NCT05238922) [65,69,70,71].

Although no preclinical studies in melanoma have been published to date evaluating the efficacy of a pan-CDK2/4/6 inhibitor or the combination of a CDK4/6 inhibitor with a CDK2 inhibitor, studies have explored the impact of CDK2 inhibition on melanoma growth control [68]. Yang et al. demonstrated that directly targeting CDK2 activity led to growth arrest, induced apoptosis, and suppressed the migration of melanoma cell lines [68]. Additional studies have demonstrated a role for CDK2 activity as a mechanism of resistance to CDK4/6is in melanoma [53,72]. Two studies revealed that melanomas resistant to CDK4/6i treatment exhibited higher levels of MDM2 and MDM4 proteins compared to treatment-sensitive cells. This suggests a potential role for these proteins in mediating resistance mechanisms, highlighting them as possible therapeutic targets to overcome CDK4/6i resistance [72,73]. When CDK4/6 inhibitors like palbociclib or ribociclib were combined with an MDM2 antagonist such as nutlin 3a, p53 levels were preserved, allowing for the increased expression of p21CIP1, the decreased phosphorylation of RB, and cellular growth arrest [72,73]. Taken together, these studies suggest that combining CDK4/6is with direct CDK2 inhibition (instead of relying on MDM2 antagonists to increase p53 and p21 levels) could further enhance CDK4/6i therapeutic efficacy by potentially overcoming CDK4/6i resistance mechanisms in melanoma treatment.

Another strategy to indirectly target CDK2 via p21 upregulation involves inhibiting PRMT5. PRMT5 methylates spliceosome factors that regulate MDM4, a negative p53 regulator. PRMT5 inhibition lowers MDM4 levels, resulting in increased levels of p53 and p21, which in turn inhibits CDK2 [53]. Cyclin D1–CDK4 also activates PRMT5 via MEP50 phosphorylation [74]. The dual inhibition of CDK4/6 (palbociclib) and PRMT5 (GSK3326595) more effectively suppressed tumor growth than palbociclib alone in melanoma models with acquired resistance to palbociclib [53].

## 8. Combining PI3K/Akt/mTOR and CDK4/6 Inhibitors for Enhanced Cancer Therapy

Another promising strategy to increase the effectiveness of CDK4/6i treatment is to combine one of these agents with drugs that target mTOR. The PI3K/Akt/mTOR signaling pathway promotes the expression of Cyclin D, and AKT regulates mTOR (mammalian target of rapamycin) through both direct and indirect phosphorylation pathways [36,75]. Directly, AKT can phosphorylate mTOR itself, enhancing its activity within the mTORC1 complex to promote protein synthesis and cell survival. Indirectly, AKT influences mTOR by phosphorylating TSC2 (Tuberous Sclerosis Complex 2), an inhibitor of the GTPase Rheb, which is essential for mTORC1 activation. When AKT phosphorylates TSC2, it removes its inhibitory effects on Rheb, allowing Rheb to activate mTORC1. These combined pathways underscore the dual regulatory role of AKT in mTOR signaling, highlighting its critical impact on cellular growth and the cell cycle [76] (Figure 1).

CDK4/6 interacts with this pathway by phosphorylating TSC2, similar to the function of AKT [77]. As described above, the phosphorylation of TSC2 leads to the activation of mTORC1, a critical regulator of cell growth and metabolism, particularly in cancer cells. This recently described role for CDK4/6 underscores its broader impact on tumorigenic pathways beyond RB phosphorylation. Therefore, inhibiting CDK4/6 may also disrupt the mTORC1 pathway, potentially amplifying the antitumor effects of CDK4/6is [77]. Studies assessing the mechanisms of acquired resistance in palbociclib-resistant cell lines and xenografts found that combined CDK4/6 and mTOR inhibition could prevent the downstream pathway reactivation of the PI3K/AKT/mTOR pathway [78]. Consequently, co-targeting mTOR with CDK4/6 inhibitors has been explored in cancers including breast cancer. Some breast cancers may have active mTORC1 signaling, where the modulation of TSC2 activity may offer additional therapeutic benefits [78,79,80,81]. Similarly, models of HER2+ breast cancer have also found that CDK4/6 inhibition suppresses TSC2 phosphorylation and decreases mTORC1 activity [82,83]. There are clinical trials in breast cancer testing hormone therapy combined with CDK4/6 inhibitors and mTOR inhibitors (ex. NCT02871791, NCT02732119).

In melanoma, three studies that explored resistance to combined CDK4/6is and MEK inhibition found that resistant tumors were frequently characterized by the activation of the PI3K pathway [84,85,86]. This pathway was often upregulated via the phosphorylation and activation of ribosomal protein S6 kinase (S6K) or the amino acid transporter SLC36A1 [84,85,86]. The phosphorylation of S6K, which contributes to cellular growth control and metabolism, is regulated by the mTOR pathway. The elevated phospho-S6K levels in CDK4/6i plus MEKi-resistant tumors underscore a therapeutic opportunity for mTORC1/2 inhibition with agents such as AZD2014. Notably, combining an mTOR inhibitor or, more specifically, an S6 kinase inhibitor has been shown to overcome this resistance [84,87]. Similarly, Yoshida et al. demonstrated that mTORC1 activation drives resistance to CDK4/6 inhibitors, further supporting the rationale for the dual targeting of these pathways. They found that combining mTORC1 inhibition (using everolimus) with CDK4/6is (using palbociclib) increased therapeutic efficacy compared to monotherapies [85]. Consequently, these findings suggest that incorporating mTORC1/2 inhibitors could effectively overcome resistance to both MEK and CDK4/6 inhibitors, potentially improving treatment outcomes.

## 9. Immunotherapy in Combination with CDK4/6 Inhibitors

Immune checkpoint blockade (ICB) has improved clinical outcomes in many cancers, perhaps none more dramatically than melanoma. In the CheckMate 067 trial that treated patients with unrescectable advanced and/or metastatic melanoma, the median overall survival using the combination anti-PD1 (nivolumab) plus anti-CTLA4 (ipilimumab) was 71.9 months compared with 36.9 months using nivolumab monotherapy or 19.9 months using ipilimumab monotherapy [88,89]. Despite these advances, a notable proportion of patients fail to respond to immunotherapy, emphasizing the need for continued research on overcoming treatment resistance.

CDK4/6 inhibition can modulate immune responses in both T cells and tumor cells, ultimately enhancing antitumor immunity. A key effect of CDK4/6 inhibition is its suppression of the nuclear factor of activated T cell (NFAT) transcription factors, which play a central role in regulating T-cell activation [90]. By inhibiting NFAT, CDK4/6 inhibitors enhance the activation of effector T cells, increasing their antitumor response. Additionally, CDK4/6 inhibition reduces immune suppression by blocking T regulatory cell (Treg) proliferation while also promoting the expansion of CD8+ cytotoxic T lymphocytes (CD8+ T cells) [91,92,93]. Furthermore, CDK4/6 inhibition alters the differentiation of cytotoxic T lymphocytes, imparting them with memory-like characteristics that help sustain ongoing antitumor immunity [91,93].

CDK4/6 inhibition enhances tumor immunogenicity by upregulating MHC class I expression on tumor cells, which can improve immune recognition [94]. CDK4/6 inhibitors can also elevate PD-L1 expression by preventing its degradation through the Cullin 3–SPOP E3 ligase pathway. Under normal conditions, CDK4 facilitates the phosphorylation of SPOP, an adaptor protein of the Cullin 3 complex that binds to PD-L1, triggering PD-L1’s degradation via the proteasome. Treatment with palbociclib reduces SPOP phosphorylation, which accelerates SPOP degradation, thereby reducing PD-L1 degradation [95,96]. Increased PD-L1 expression can help cancer cells evade immune detection, but increased expression is also correlated with favorable response to anti-PD-1-based immunotherapies [88]. Combining CDK4/6 inhibitor treatment with anti-PD-1 immunotherapy enhances tumor regression and improves overall survival in immune-proficient mice bearing colorectal CT26 tumors [95,97,98]. This strategy (CDK4/6 inhibition with immunotherapy) is currently being explored in melanoma in the PLATforM clinical trial (NCT03484923).

MEKi and CDK4/6i combination therapy was found to increase the capacity of melanoma cells to present antigens and recruit cytotoxic CD8+ T cells to the tumor [99]. On the other hand, Lelliott et al. demonstrated in mice that adding a BRAFi to the combination of a CDK4/6i and MEKi provides greater efficacy compared to the combination of a CDK4/6i and MEKi alone but resulted in tumors that were unresponsive to the subsequent dual immune checkpoint blockade with anti-CTLA4 and anti-PD1 antibodies. They found that the triple combination depleted immune-potentiating myeloid populations such as proinflammatory macrophages and cross-priming CD103+ dendritic cells in melanoma [92]. Thus, while triple therapy may have beneficial antitumor intrinsic properties to control melanoma tumor growth, it may decrease tumor susceptibility to ICB. However, another study from the same investigators found that BRAF-MEK-CDK4/6i combined with adoptive cell transfer (ACT) led to a sustained antitumor response in a BRAFi-sensitive murine model of melanoma [100]. Thus, future studies need to identify the optimal CDK4/6i-based combinations and timing with other targeted therapies to effectively modulate the tumor immune microenvironment to promote tumor killing.

## 10. Other Potential Combination Treatments with CDK4/6is

Besides the direct cell cycle inhibitory and possible immune modulating effects noted above, the combination of CDK4/6is with MEKis was shown to increase reactive oxygen species (ROS) levels within cancer cells. Elevated ROS can induce oxidative stress, damaging cellular components and potentially leading to cell death. This outcome has sparked interest in studies aimed at further amplifying ROS as a therapeutic strategy. By promoting oxidative stress beyond the cancer cell’s tolerance threshold, researchers hope to enhance the efficacy of dual CDK4/6i and MEKi therapies, potentially providing a powerful approach to targeting resistant or aggressive cancers.

A recent study using the single-cell RNA sequencing of NRAS-mutant melanoma treated with CDK4/6is and MEK1/2is found that the activation of the ATP-gated ion channel P2RX7 can be a marker for response to such combination therapy [101]. P2RX7 is known to mediate Ca^2+^ influx, increasing the production of ROS. The activation of P2RX7 with BzATP, a P2RX7 small-molecule agonist, further induces high intracellular levels of ROS induced by CDK4/6 and MEK co-inhibition. Additionally, the high expression of P2RX7 is associated with a longer overall and progression-free survival in NRAS-mutant melanoma patients but not in BRAF-mutant melanoma patients, suggesting that specific subsets of patients might be responsive to P2RX7 activation [101]. Another study found that a P2RX7 antagonist increased immune infiltration in melanoma and could be a therapeutic approach in melanoma [102]. Thus, the question of whether to activate or inactivate PRX7 in combination with CDK4/6 inhibition remains to be further elucidated.

## 11. Conclusions

The dysregulation of the Cyclin D–CDK4/6 pathway is a hallmark of melanoma progression, making it an attractive target for intervention. CDK4/6is are the standard of care when used in combination with other agents in breast cancer, so a key challenge in melanoma is identifying molecularly defined subsets of melanoma that may similarly respond to appropriate combination therapies. Among the combination therapies explored, combining CDK4/6is with BRAF and MEK inhibitors (or MEK inhibitors only for BRAF wild-type tumors) has had limited efficacy and high toxicity. This underscores the complex nature of melanoma’s signaling networks and highlights the need to look beyond the MAPK pathway for effective combination treatment strategies. Preclinical data suggest exploring CDK4/6i combinations with other promising therapies in phase 1/2 trials. Identifying biomarkers, such as elevated CDK6, CDK2, or MDM2/4 levels, may refine patient selection and improve outcomes. Additionally, the relative and absolute expression levels of CDK4 and CDK6, along with factors within the tumor microenvironment, may offer further insights into treatment responsiveness. This evolving therapeutic landscape emphasizes the need for continued research into the molecular determinants of sensitivity and resistance, as well as the development of innovative combination therapies that exploit the vulnerabilities of melanoma cells. By addressing these challenges, targeting the Cyclin D–CDK4/6 axis has the potential to play a pivotal role in advancing melanoma treatment and improving patient outcomes.

## Figures and Tables

**Figure 1 cancers-17-01291-f001:**
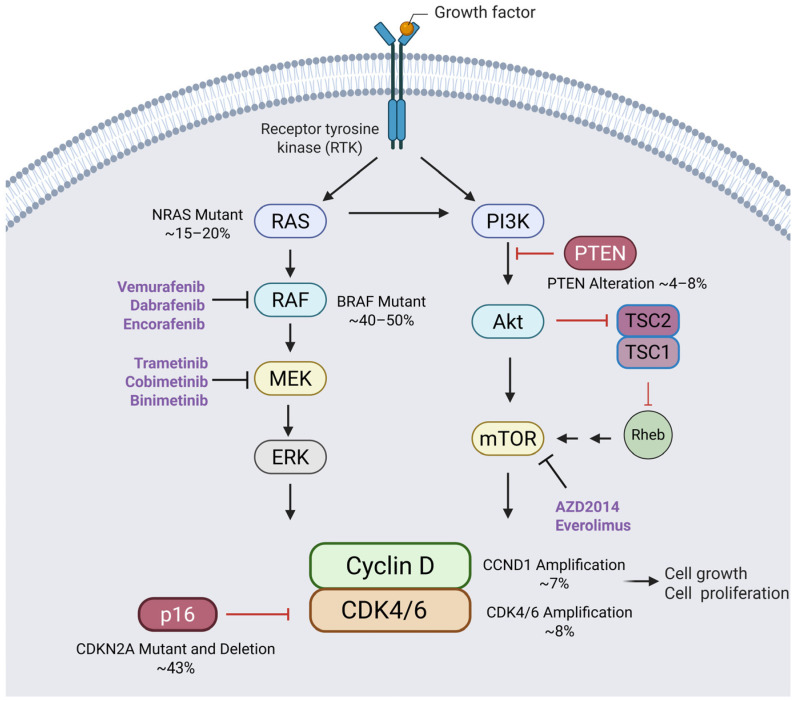
**Cell cycle activation in melanoma via the RAS-RAF-MEK-ERK and PI3K-Akt-mTOR pathways.** Cell cycle progression in melanoma is primarily driven by the RAS-RAF-MEK-ERK and PI3K-Akt-mTOR signaling pathways. These pathways regulate key Cyclin-dependent kinases (CDK4/6), promoting the transition from the G1 to S phase. The left-hand side depicts the RAS-RAF-MEK-ERK pathway. The activation of RAS leads to the phosphorylation and activation of RAF, MEK, and ERK, resulting in the upregulation of Cyclin D. Cyclin D binds to CDK4/6, triggering the phosphorylation of the retinoblastoma (RB) protein (depicted in Figure 2). The right-hand side depicts the PI3K-Akt-mTOR pathway and TSC2 regulation. The TSC1-TSC2 complex functions as a GTPase-activating protein (GAP), preventing Rheb from activating mTORC1. This suppresses protein synthesis and cell growth. Key tumor suppressors altered in melanoma, such as PTEN and p16, regulate these pathways. PTEN antagonizes PI3K-Akt signaling, while p16 inhibits Cyclin D–CDK4/6 activity, ensuring tight control over the cell cycle. Therapeutic inhibitors targeting these pathways are shown (highlighted in purple). This figure was generated using BioRender (https://www.biorender.com/).

**Figure 2 cancers-17-01291-f002:**
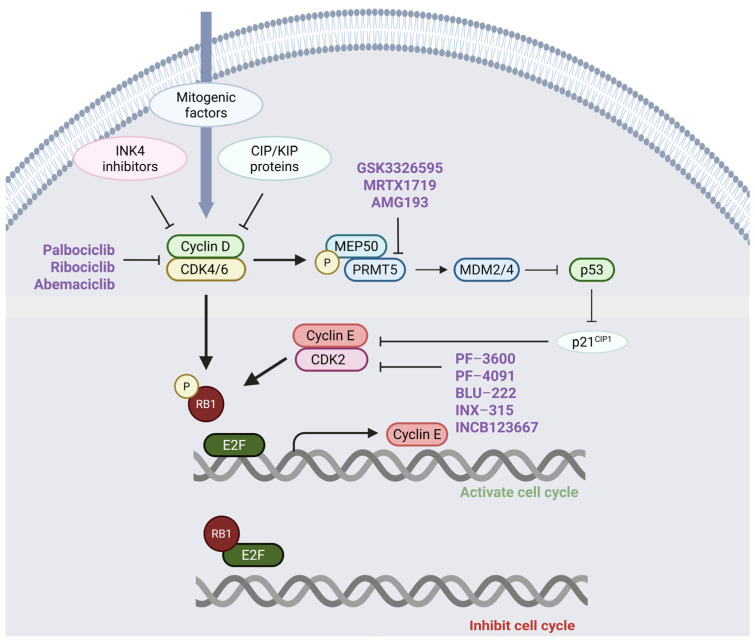
**The Cyclin D–CDK4/6–RB–E2F pathway and its interaction with the MEP50/PRMT5-p53-p21-CDK2 axis.** The activity of Cyclin D–CDK4/6 is controlled by INK4 inhibitors and CIP/KIP proteins, which regulate cell cycle progression. Cyclin D–CDK4/6 complexes phosphorylate RB, which releases E2F transcription factors, enhancing Cyclin E transcription. Cyclin E subsequently binds and activates CDK2, resulting in RB hyperphosphorylation and the transcription of genes that drive the cell division cycle. MEP50, a substrate of Cyclin D–CDK4/6 complexes, increases the activity of PRMT5, which can reduce the expression of p21 (a CIP/KIP family member) by enhancing the expression of MDM4, which normally inhibits the activity of p53. Reduced p21 expression leads to the diminished inhibition of CDK2, facilitating cell cycle progression. Several inhibitors are either developed or in development to target various components of this pathway, including CDK4/6, CDK2, and PRMT5 (highlighted in purple). This figure was generated using BioRender (https://www.biorender.com/).

**Table 1 cancers-17-01291-t001:** Current and completed clinical trials of CDK4/6 inhibitors in melanoma: a comprehensive review of abemaciclib, palbociclib, and ribociclib from ClinicalTrials.gov.

Clinical Trial Identifier	Study Name	Phase/Recruitment Status	Last Update Posted	Therapies
CDK4/6 inhibition alone				
NCT02308020	A Study of Abemaciclib (LY2835219) in Participants with Breast Cancer, Non-small Cell Lung Cancer, or Melanoma That Has Spread to the Brain	Phase 2; Completed	19 December 2020	CDK4/6i (abemaciclib)
NCT01037790	Phase II Trial of the Cyclin-Dependent Kinase Inhibitor PD 0332991 in Patients with Cancer	Phase 2; Completed	11 March 2021	CDK4/6i (palbociclib)
NCT05063058	Biomarker-driven Therapy for Melanoma (TREAT20plus)	Phase N/A; Completed	30 September 2021	CDK4/6i (palbociclib)
NCT01394016	A Phase 1 Study of LY2835219 In Participants With Advanced Cancer	Phase 1; Completed	19 September 2024	CDK4/6i (abemaciclib)
NCT03310879	Study of the CDK4/6 Inhibitor Abemaciclib in Solid Tumors Harboring Genetic Alterations in Genes Encoding D-type Cyclins or Amplification of CDK4 or CDK6	Recruiting; Phase 2	24 January 2025	CDK4/6i (abemaciclib)
NCT02465060	Targeted Therapy Directed by Genetic Testing in Treating Patients with Advanced Refractory Solid Tumors, Lymphomas, or Multiple Myeloma (The MATCH Screening Trial)	Phase 2; Active, not recruiting	09 April 2025	CDK4/6i (palbociclib)
CDK4/6 inhibition + MAPK inhibition				
NCT02065063	A Study to Investigate the Safety, Pharmacokinetics, Pharmacodynamics, and Anti-Cancer Activity of Trametinib in Combination with Palbociclib in Subjects With Solid Tumors	Phase 1; Completed	1 June 2018	CDK4/6i (palbociclib) and MEKi (trametinib)
NCT01781572	A Phase Ib/II Study of LEE011 in Combination With MEK162 in Patients With NRAS Mutant Melanoma	Phases 1 and 2; Completed	7 December 2020	CDK4/6i (ribociclib) and MEKi (MEK162)
NCT02857270	A Study of LY3214996 Administered Alone or in Combination with Other Agents in Participants With Advanced/Metastatic Cancer	Phase 1; Completed	22 November 2022	CDK4/6i (abemaciclib) and ERK1/2i (temuterkib)
NCT02159066 (LOGIC-2)	LGX818 and MEK162 in Combination with a Third Agent (BKM120, LEE011, BGJ398 or INC280) in Advanced BRAF Melanoma (LOGIC-2)	Phase 2; Completed	5 March 2024	CDK4/6i (ribociclib), MEKi (binimetinib), and BRAFi (encorafenib)
NCT04720768 (CELEBRATE)	Encorafenib, Binimetinib and Palbociclib in BRAF-mutant Metastatic Melanoma CELEBRATE (CELEBRATE)	Recruiting; Phase 1b	5 January 2024	CDK4/6i (palbociclib), MEKi (binimetinib), and BRAFi (encorafenib)
NCT03454035	Ulixertinib/Palbociclib in Patients with Advanced Pancreatic and Other Solid Tumors	Recruiting; Phase 1	18 March 2025	CDK4/6i (palbociclib) and ERK1/2i (ulixertinib)
NCT02645149 (MatchMel)	Molecular Profiling and Matched Targeted Therapy for Patients with Metastatic Melanoma (MatchMel)	Recruiting; Phase 2	07 January 2025	CDK4/6i (ribociclib) and MEKi (trametinib)
NCT04594005	CDK4/6 Tumor, Abemaciclib, Paclitaxel	Phases 1 and 2; Active, not recruiting	18 December 2024	CDK4/6i (abemaciclib) and paclitaxel
NCT01543698	A Phase Ib/II Study of LGX818 in Combination With MEK162 in Adult Patients With BRAF Dependent Advanced Solid Tumors	Phases 1 and 2; Completed	13 March 2024	CDK4/6i (ribociclib), MEKi (MEK162), and BRAFi (encorafenib)
NCT04417621	Study of Efficacy and Safety of LXH254 Combinations in Patients with Previously Treated Unresectable or Metastatic Melanoma	Phase 2; Active, not recruiting	04 April 2025	CDK4/6i (ribociclib) and RAFi (LXH254)
NCT02974725	A Phase Ib Study of LXH254-centric Combinations in NSCLC or Melanoma	Phase 1; Terminated	17 May 2024	CDK4/6i (ribociclib) and RAFi (LXH254)
CDK4/6 inhibition + immunotherapy				
NCT03484923 (PLATforM)	Study of Efficacy and Safety of Novel Spartalizumab Combinations in Patients with Previously Treated Unresectable or Metastatic Melanoma (PLATforM)	Phase 2; Completed	18 June 2024	CDK4/6i (ribociclib) and PD-1 receptor antibody (Spartalizumab)
NCT02791334 (PACT)	A Study of Anti-PD-L1 Checkpoint Antibody (LY3300054) Alone and in Combination in Participants with Advanced Refractory Solid Tumors (PACT)	Phase 1; Completed	27 September 2024	CDK4/6i (abemaciclib) and PD-L1 antibody (LY3300054)

Note: N/A: Not Applicable.

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
