# Peer review of "Recent Developments in Targeting the Cell Cycle in Melanoma"

_cancers, 2025, doi:10.3390/cancers17081291_

Round 1
Reviewer 1 Report
Comments and Suggestions for Authors
The abstract is too cumbersome, and the author should reorganize and simplify the description of the abstract.
The importance of targeting the CDK4/6 pathway should be stressed in the background.
Figures 1 and 2 should be further optimized by adding titles and annotations to make them easier for readers to understand.
To offer a wider context, the review could incorporate comparisons with CDK4/6 inhibitor research in other cancers, such as breast cancer and lung cancer. This would bring out the unique features of melanoma biology and treatment challenges, enriching the discussion.
In the background section, the authors should include descriptions of the cell cycle and DNA replication. The following references are suggested for the authors to add to the manuscript for discussion: PMID: 35439677 and PMID: 36202931.
Author Response
The abstract is too cumbersome, and the author should reorganize and simplify the description of the abstract.
We thank the Reviewer for their encouragement to improve the readability of the abstract. We have reorganized and simplified it per the reviewer's suggestion (Manuscript page 1).
The importance of targeting the CDK4/6 pathway should be stressed in the background.
We thank the Reviewer for helping us more tightly focus the Background section. In response, we revised the first paragraph to highlight the critical role of the CDK4/6 pathway in cell cycle regulation and underscore the importance of targeting it for cancer therapy (Manuscript line 37-44).
Figures 1 and 2 should be further optimized by adding titles and annotations to make them easier for readers to understand.
We thank the Reviewer for their suggestion to improve Figures 1 & 2. We edited the legends for improved readability (Manuscript lines 75-87; 115-126).
To offer a wider context, the review could incorporate comparisons with CDK4/6 inhibitor research in other cancers, such as breast cancer and lung cancer. This would bring out the unique features of melanoma biology and treatment challenges, enriching the discussion.
We thank the Reviewer for this valuable suggestion. We added discussion of other cancers in several places throughout the manuscript including Sections 4, 6 and 8. Specifically, the new text briefly discusses studies in breast and lung cancers, as well as gastrointestinal stromal tumors, T-ALL, and glioblastoma. We also more specifically discuss the findings in other cancers throughout the manuscript in places where we previously simply listed references referring to those studies.
In the background section, the authors should include descriptions of the cell cycle and DNA replication. The following references are suggested for the authors to add to the manuscript for discussion: PMID: 35439677 and PMID: 36202931.
We thank the Reviewer for suggesting that we specifically mention cell cycle and DNA replication in the Background section. We have made those edits, which provide better context for our review (Manuscript lines 38 - 42).
Reviewer 2 Report
Comments and Suggestions for Authors
This review article provides an overview on Cyclin/CDK cell-cycle regulation with an emphasis on cell-cycle dysregulation and use of CDK4/6 inhibitors in melanoma. The authors highlight some clinical trials using CDK4/6 inhibitors and then go into a detailed mechanistic overview of why current trials have not been as promising as expected, with a focus on CDK2 compensation underlying this lack of response.
Strengths:
-With several pharmaceutical companies currently developing and testing CDK4/6, CDK4, CDK2, and CDK2/4/6 inhibitors, and several ongoing clinical trials testing these drugs, a review focusing on CDK4/6 inhibitors and their efficacy, or lack therefore, is extremely important for the field.
-The authors are versed in the literature of cell-cycle regulation and dysregulation.
-The authors provided a significant and comprehensive background on the cell-cycle and how different signalling pathways altered in melanoma (MAPK/PI3K) feed into the cell cycle.
Weaknesses:
-There is almost no literature review on the preclinical in vivo or in vitro work that supports the use of CDK4/6 inhibitors in melanoma, other than discussing genetics. Also, when discussing common alterations in the Cyclin/CDK axis in melanoma (e.g. CDKN2A loss, CCND1 amplification, CDK4 copy gains/alterations) it is unclear if these are seen at initial biopsy or develop as resistance mechanisms to first-line therapy. This is important because this influences if CDK4/6i + MAPKi/immunotherapy would be good as a first line therapy or in advanced stages after resistance to first line therapy.
-Many trials listed look like they were done in advanced stages after first line therapy which is also important to discuss as this may explain the poor clinical response observed.
-Paper reads as a pull for using CDK2/4/6 inhibitors as an alternative to ineffective CDK4/6 inhibitors. However, this overlooks a big limitation for why many CDK4/6+MAPKi trials have not had the best outcomes: patient stratification. In the one trial where they stratified patients by cell-cycle related alterations, patients had a much greater response. The authors should discuss that clinicians need to consider if melanoma patients have p16 mutations, CCDN1 amplification, CDK4/6 alterations because this is the population most likely to experience increased efficacy, particularly as first line treatment. There are several preclinical papers highlighting a role for p16 mutations, for example, in sensitizing melanoma lines and mouse models to combination CDK4/6 + MAPKi, that were not discussed.
-Overview of previously performed and on-going clinical trials is confusing
- Should separate trials list according to drug treatments: CDK4/6i alone, combo CDK4/6i+MAPKi (first-line, then second line therapy) and then CDK4/6i+immunotherapy.
- Some trials discussed in the text are not in the table (NCT01543698, NCT01781572) and several completed trials for MAPKi+CDK4/6i that are in the table are not mentioned or even cited in the text when discussing efficacy of MAPKi+CDK4/6i (NCT02065063, NCT02159066 (LOGIC-2), NCT02857270).
- Response rate is confusing as a metric for determining if treatment is effective or not, making it difficult to compare trials. For example, with single agent abemacicilib, the disease control rate was 27% vs with ribociclib and binimetinib it was 19%. MAPKi is a standard of care so it's confusing that the response rate is worse in the combo than the single agent alone. Is response rate a good measure for this? Would PFS or some other measure better capture outcomes? It would also be beneficial to clarify the standard of care in all of these situations and what the response rate or PFS is for the standard so they can be accurately compared to the results of these new trials.
-In section 5/6, many of the papers discussed about p21/p27 binding to CDK4/6 and therefore changing p21/p27 binding to CDK2 were not done in melanoma lines. While this doesn’t mean this could not happen in these contexts and explain melanoma resistance to CDK 4/6i, there is a significant amount of detail (a paragraph per paper) describing each of these papers, when a few sentences total would suffice. This is also why the paper reads as overly focused on p21/p27/CDK2 when things like patient stratification, first-line vs second-line therapy, etc are all important reasons to consider why these trials may not have been as efficacious as expected.
- At the end of the review, the authors start discussing pre-clinical studies for why CDK4/6i + MAPKi might not be effective, but more literature review on papers done specifically in melanoma lines is needed. This part reads more as a review on CDK4/6i in a general context vs in melanoma alone.
Literature coverage:
-Authors need more preclinical studies explaining why CDK4/6 inhibitors were even considered for clinical trials as well as well as discussing lack of response or resistance to combination CDK4/6i+MAPKi/immunotherapy
-The restriction point is not discussed as a concept, even though it is a central concept in this area
Writing style and clarity:
-Manuscript writing itself could be improved in terms of typos/spelling mistakes and switching between lower and upper case (ex, Cyclin vs cyclin).
-Paragraphs that only cite a single paper should be condensed into a few sentences to streamline and improve readability and flow.
-Sherr gets mentioned with no first name, unlike other early pioneers
-line 73, Cyclin E does not bind CDK4/6
-line 93, Cyclin A gets brought up with no mention of its main role as a CDK2 activator
-line 97, ‘after mitosis, Rb returns to its dephoshorylated state’ - The cell cycle field has shifted toward the view that under optimal growth conditions, a large fraction of cells do not dephosphorylate Rb and instead remain committed to completing another cell cycle.
-line 151, the field has shifted toward the view that CDK4/6 inhibitors arrest cells in G0, or immediately before the R point.
-line 127, “R phosphorylation” typo.
-line 197, what is a molecular subset?
-line 204-205 – that sentence should be referenced; is that claim coming from the CDK4/6 knockout mouse work?
-line 231, reference 47 is mis-cited. Reference 47 should instead be discussed in the CDK2 section around line 300.
-lines 335-350, why is there so much discussion of PRMT5? This feels unbalanced.
-lines 364-369, the idea that CDK4/6 activates mTorC1 by phosphorylation of TSC2 is interesting. But there are other papers besides ref 71 that show this, no?
Author Response
This review article provides an overview on Cyclin/CDK cell-cycle regulation with an emphasis on cell-cycle dysregulation and use of CDK4/6 inhibitors in melanoma. The authors highlight some clinical trials using CDK4/6 inhibitors and then go into a detailed mechanistic overview of why current trials have not been as promising as expected, with a focus on CDK2 compensation underlying this lack of response.
There is almost no literature review on the preclinical in vivo or in vitro work that supports the use of CDK4/6 inhibitors in melanoma, other than discussing genetics. Also, when discussing common alterations in the Cyclin/CDK axis in melanoma (e.g. CDKN2A loss, CCND1 amplification, CDK4 copy gains/alterations) it is unclear if these are seen at initial biopsy or develop as resistance mechanisms to first-line therapy. This is important because this influences if CDK4/6i + MAPKi/immunotherapy would be good as a first line therapy or in advanced stages after resistance to first line therapy.
We thank the Reviewer for their concern that the primary rationale for the use of CDK4/6 inhibitors in melanoma is based on the relatively high frequency of genetic alterations to the cell cycle control circuitry, namely Cyclin D, CDK4 and CDKN2A. We did note in the Introduction that the established efficacy of CDK4/6 inhibitors in other cancers such as hormone receptor-positive (HR+) breast cancer also supports their potential use in melanoma. In response to the Reviewer’s comment, we’ve added additional preclinical references that focused on using CDK4/6 inhibitors as monotherapy or as combination therapies with MEK inhibitors in an attempt to overcome acquired resistance to BRAF inhibitors (Lines 162 - 177).
Regarding the issue of whether common alterations in the Cyclin/CDK axis occur at initial biopsy or develop as resistance mechanisms to first-line therapy, we agree with the Reviewer that alterations to the Cyclin/CDK axis may occur in primary melanomas, or later in metastatic tumors as part of tumor progression. Much of the literature describing these alterations is based on studies of metastatic tumors and their derived cell lines that were collected prior to the widespread use of targeted and immunotherapies in melanoma. Regarding the potential use of CDK4/6i + MAPK/immunotherapy, such a trial would have to be initiated in the second- or third-line setting because of the limited efficacy of CDK4/6i’s in melanoma to date, and the availability of several FDA-approved first-line options with proven efficacy (e.g., combination or single agent immunotherapies, and combination BRAF-MEK targeted therapies).
Many trials listed look like they were done in advanced stages after first line therapy which is also important to discuss as this may explain the poor clinical response observed.
We thank the Reviewer for this observation and suggestion to include the relevant discussion in the manuscript. We inserted additional text to address this point (Manuscript lines 194 - 200).
Paper reads as a pull for using CDK2/4/6 inhibitors as an alternative to ineffective CDK4/6 inhibitors. However, this overlooks a big limitation for why many CDK4/6+MAPKi trials have not had the best outcomes: patient stratification. In the one trial where they stratified patients by cell-cycle related alterations, patients had a much greater response. The authors should discuss that clinicians need to consider if melanoma patients have p16 mutations, CCDN1 amplification, CDK4/6 alterations because this is the population most likely to experience increased efficacy, particularly as first line treatment. There are several preclinical papers highlighting a role for p16 mutations, for example, in sensitizing melanoma lines and mouse models to combination CDK4/6 + MAPKi, that were not discussed.
We thank the Reviewer for this comment and agree that selecting patients for treatment with CDK4/6i + MAPKi treatment based on alterations to the Cyclin/CDK axis is theoretically appealing based on preclinical studies. We added text describing a large study in acral melanoma suggesting that tumors with alterations in the Cyclin D-CDK4/6-CDKN2A axis would be responsive to palbociclib (Lines 164 – 168) (Kong Y et al (2017) Clin Ca Res). However, that strategy failed in the clinic (Lines 184 – 187) (Mao L et al (2021) Eur J Cancer). In non-acral cutaneous melanoma the strategy appeared to show some promise based on retrospective analyses of data from the Phase Ib/II study of ribociclib plus binimetinib in NRASmutant melanoma (Shuler et al). In that study, the subset of patients with alterations in the Cyclin/CDK axis experienced an overall response rate (ORR) of 32.5%, which was higher than the 10% ORR in patients lacking an alteration. However, the 32.5% ORR is not particularly encouraging in the context of other experimental therapies in the second- or third-line setting for melanoma. Importantly, the main limitation of combining CDK4/6i and MAPKi in this and similar studies appears to be toxicity. Specifically, the inability to adequately dose CDK4/6i’s at levels comparable to breast cancer. In the Shuler et al study, the ribociclib dose in the Phase II segment was 200mg daily, which is only one-third of the 600mg daily dose approved for breast cancer, and the plasma concentration time curve over the dosing interval was only 10% of that for the breast cancer dose. We emphasize this issue in lines 212-214.
Additionally, a very recent report from the NCI-MATCH trial demonstrated that treating patients whose solid tumors or lymphomas had at least 7 copies of CDK4 or CDK6 with single-agent palbociclib was of limited efficacy casting doubt on the utility of using genomic amplification of CDK4 or CDK6 as a biomarker to stratify patients for CDK4/6i treatment (Manuscript lines 188-194).
Overview of previously performed and on-going clinical trials is confusing. Should separate trials list according to drug treatments: CDK4/6i alone, combo CDK4/6i+MAPKi (first-line, then second-line therapy) and then CDK4/6i+immunotherapy.
We thank the Reviewer for this suggestion to improve the clarity of Table 1. In the revised manuscript, we reclassified the clinical trials into subgroups: CDK4/6i alone, combination therapy with CDK4/6i and MAPKi (first line followed by second-line therapy), and CDK4/6i with immunotherapy (Table 1, Manuscript line 245).
Some trials discussed in the text are not in the table (NCT01543698, NCT01781572) and several completed trials for MAPKi+CDK4/6i that are in the table are not mentioned or even cited in the text when discussing efficacy of MAPKi+CDK4/6i (NCT02065063, NCT02159066 (LOGIC-2), NCT02857270).
We thank the Reviewer for the thorough review and feedback. In the revised manuscript we expanded Table 1 and the text to include the information mentioned (Manuscript lines 225-243, and Table 1 beginning at lines 245).
Response rate is confusing as a metric for determining if treatment is effective or not, making it difficult to compare trials. For example, with single agent abemacicilib, the disease control rate was 27% vs with ribociclib and binimetinib it was 19%. MAPKi is a standard of care so it's confusing that the response rate is worse in the combo than the single agent alone. Is response rate a good measure for this? Would PFS or some other measure better capture outcomes? It would also be beneficial to clarify the standard of care in all of these situations and what the response rate or PFS is for the standard so they can be accurately compared to the results of these new trials.
We thank the Reviewer for this question and comment about response rate (RR) and progression-free survival (PFS) as measures of the effectiveness of experimental therapeutics, and the challenges of making comparisons across clinical trials. Both RR and PFS are important endpoints in cancer clinical trials. In the context of phase II trials of CDK4/6i, RR is often a preferred endpoint because it is a direct assessment of whether a therapy is actively shrinking tumors, and it is relatively easy to measure radiographically according to standardized criteria (RECIST - Response Evaluation Criteria in Solid Tumors). Disease control rate (DCR) is a less desirable endpoint because it includes patients with stable disease (in addition to complete responders and partial responders), and this could be due to a coincidental slowing of disease progression and not to the effect of the experimental therapy. The Reviewer is also asking for a direct cross-trial comparison of DCRs in the ribociclib plus binimetinib trial (NCT01781572) versus the abemacicilib monotherapy trial (NCT01394016). Such comparisons can often be misleading due to differences in patient populations, sample sizes and prior treatments. In this case there are several differences between these trials, not only the addition of binimetinib treatment in one trial, but also the differences in the CDK4/6i’s employed. Ribociclib and abemacicilib are different drugs with differing levels of activity against CDK4 and CDK6. Also, as noted above, the addition of the MEK inhibitor binimetinib increases the toxicity of the regimen, which can also limit the effectiveness of the regimen resulting in differing rates of patients discontinuing treatment om the different trials. Finally, NCT01781572 specifically enrolled patients with NRAS-mutant melanoma, whereas NCT01394016 included all subtypes of advanced melanoma, which may have contributed to differences in treatment outcomes. Taken together, there are several reasons why one should exercise caution in trying to compare the DCRs between these trials.
In section 5/6, many of the papers discussed p21/p27 binding to CDK4/6 and therefore changing p21/p27 binding to CDK2 were not done in melanoma lines. While this doesn’t mean this could not happen in these contexts and explain melanoma resistance to CDK 4/6i, there is a significant amount of detail (a paragraph per paper) describing each of these papers, when a few sentences total would suffice. This is also why the paper reads as overly focused on p21/p27/CDK2 when things like patient stratification, first-line vs second-line therapy, etc are all important reasons to consider why these trials may not have been as efficacious as expected.
We thank the Reviewer for pointing out an overemphasis on non-melanoma data with respect to p21/p27 and CDK2. In response we condensed the discussion of these studies into a few sentences while ensuring the key mechanistic insights remain clear (Section 5 paragraph beginning at line 256, and Section 6, paragraph beginning at line 277).
At the end of the review, the authors start discussing pre-clinical studies for why CDK4/6i + MAPKi might not be effective, but more literature review on papers done specifically in melanoma lines is needed. This part reads more as a review on CDK4/6i in a general context vs in melanoma alone.
We believe the Reviewer is referring to the concluding paragraph, where we discussed the limited efficacy and high toxicity of the CDK4/6i + MAPKi combination in clinical trials. The reviewer is suggesting that we increase the review of papers based on cell line and mouse studies; however, these preclinical studies do not capture toxicity, so we do not believe that additional emphasis on cell line studies is warranted. As noted above, we expanded our discussion of preclinical melanoma studies supporting various therapeutic combinations of CDK4/6i’s with other agents to enhance effectiveness. Additionally, we included factors that may contribute to poor clinical responses, such as pre-existing tumor resistance mechanisms or the acquisition of additional mutations and emphasized the need to identify new combination approaches to improve effectiveness of CDK4/6i’s in melanoma.
Literature coverage:
Authors need more preclinical studies explaining why CDK4/6 inhibitors were even considered for clinical trials as well as discussing lack of response or resistance to combination CDK4/6i+MAPKi/immunotherapy.
We thank the Reviewer for this suggestion to add more studies to the manuscript. In response, we expanded the manuscript with a discussion of additional preclinical studies to explain the rationale behind considering CDK4/6 inhibitors for clinical trials. The additions are mostly in Section 4. We also modified the title of section 4 to reflect the additional text. It now reads, “Preclinical and Clinical Studies in Melanoma Utilizing CDK4/6 Inhibitors” (Section 4, line 156 - 178).
With respect to the comment regarding CDK4/6i in combination with MAPK inhibitors and/or immunotherapy, lack of efficacy has been observed in several trials that we reviewed. We discussed the main challenge to combining CDK4/6i with BRAF inhibitors (BRAFi) and/or MEK inhibitors (MEKi) is significant toxicity, which often leads to dose reductions or treatment discontinuation. This inability to achieve an optimal therapeutic dose in combination regimens complicates their potential clinical utility. As a result, there is an urgent need to identify novel drug combinations that can maintain therapeutic efficacy while minimizing toxicity, thereby improving patient outcomes and expanding treatment options. (Manuscript page 6, line 224-242). Combinations of CDK4/6i with immunotherapy including ongoing clinical studies is discussed in Section 9.
The restriction point is not discussed as a concept, even though it is a central concept in this area
We thank the Reviewer for pointing out this oversight. In response we incorporated an explanation for the restriction point into the manuscript (Line 99): "The stage within the G1 phase where the cell becomes irreversibly committed to the S phase is known as the restriction point, a critical checkpoint at which cells no longer require mitogenic stimuli to proceed into S phase"
Writing style and clarity:
Manuscript writing itself could be improved in terms of typos/spelling mistakes and switching between lower and upper case (ex, Cyclin vs cyclin).
Thank you for highlighting the inconsistency in capitalization. We've corrected it accordingly together with typos/spelling mistakes.
Paragraphs that only cite a single paper should be condensed into a few sentences to streamline and improve readability and flow.
Thank you for this suggestion. We agree and have removed or condensed these sections to enhance the overall readability of the manuscript.
Sherr gets mentioned with no first name, unlike other early pioneers
Thank you. We made the appropriate correction.
line 73, Cyclin E does not bind CDK4/6
Thank you for identifying this error in Figure 1 Legend. We made the correction as part of our revision to the Legend, which begins on line 80.
line 93, Cyclin A gets brought up with no mention of its main role as a CDK2 activator
Thank you for identifying this oversight. We highlighted its role as a CDK2 activator in the manuscript (Manuscript page 3, line 95).
line 97, ‘after mitosis, Rb returns to its dephoshorylated state’ - The cell cycle field has shifted toward the view that under optimal growth conditions, a large fraction of cells do not dephosphorylate Rb and instead remain committed to completing another cell cycle.
We thank the Reviewer for this comment, and agree that traditionally, Rb was thought to be dephosphorylated at the end of mitosis, resetting cell cycle control and ensuring proper checkpoint regulation in G1. This dephosphorylation was considered essential for cells to re-enter a quiescent or growth-sensitive state. However, more recent studies suggest that under proliferative conditions, many cells bypass this dephosphorylation step and instead retain phosphorylated Rb, remaining committed to continued cycling. This challenges the classical view, indicating that Rb phosphorylation status may not fully reset at mitotic exit. In response to the Reviewer’s comment and reflecting this shift in perspective, we modified the text to state that “Upon completion of mitosis, the classical view has been that RB returns to its dephosphorylated state and binds to E2F, inhibiting the expression of genes driving cell cycle progression after mitosis. However, more recent studies support the concept that under the appropriate conditions, cells can maintain phosphorylated RB and continuously proliferate” [Lines 102-106].
line 151, the field has shifted toward the view that CDK4/6 inhibitors arrest cells in G0, or immediately before the R point.
In response to this comment, the authors independently conducted new literature searches to assess the paradigm shift the Reviewer described. We found several recent studies (Rubin et al., Mol Cell. 2020, Crozier et al., EMBO J. 2022; Ding Let al., Int J Mol Sci. 2020) that described CDKs 4 and 6 as key regulators of the transition from the G1 phase to the S phase of the cell cycle. The authors of these publications highlighted that CDK4/6 inhibitors effectively suppress the proliferation of sensitive cancer cells by inducing a G1 cell cycle arrest. While some authors suggest that prolonged CDK4/6 inhibition may lead to a quiescent-like state, this state differs from true quiescence (G0). Cells arrested by CDK4/6 inhibitors remain metabolically active and can often re-enter the cell cycle upon drug withdrawal, unlike those in a genuine G0 state. Therefore, CDK4/6 inhibitors primarily arrest the cell cycle at the G1 phase, just before the restriction (R) point, rather than fully transitioning cells into G0.
line 127, “R phosphorylation” typo.
Thank you. It should have been “RB”. We corrected the text (Line 136).
line 197, what is a molecular subset?
Thank you for requesting clarification. The molecular subsets refer to the mutations or other alterations present in melanoma patient tumors that may be helpful for stratifying patients for different treatment strategies. We modified the text to “molecularly defined subsets” (Line 473).
line 204-205 – that sentence should be referenced; is that claim coming from the CDK4/6 knockout mouse work?
We thank the Reviewer for alerting us to this oversight. We added additional citations to further support our claim regarding the CDK4/6 and CDK2 redundancy based on knockout mouse models and cell line knockdown experiments (Section 5, line 250).
line 231, reference 47 is mis-cited. Reference 47 should instead be discussed in the CDK2 section around line 300.
We thank the Reviewer for their careful review. We moved reference 47 to the CDK2 section accordingly and added a second reference supporting the impact of Cyclin D overexpression on driving cell cycle progression.
lines 335-350, why is there so much discussion of PRMT5? This feels unbalanced.
We thank the Reviewer for this comment. In response, we’ve streamlined the text to improve readability (Paragraph beginning at line 349).
lines 364-369, the idea that CDK4/6 activates mTorC1 by phosphorylation of TSC2 is interesting. But there are other papers besides ref 71 that show this, no?
We thank the Reviewer for requesting additional depth on this topic. In response we've expanded the text slightly and included additional citations to better reflect the concept that CDK4/6 activates mTORC1 through TSC2 phosphorylation (Manuscript lines 373-383).
Reviewer 3 Report
Comments and Suggestions for Authors
Thank you for the opportunity to review this very interesting paper. The paper offers a well-written and comprehensive review of the current knowledge regarding the Cyclin-D-CDK4/6-RB1 signaling pathway in melanoma. This is a critical research area given the high prevalence of alterations in this pathway and potentially an interesting therapeutical target. The authors have synthesized the most relevant studies and clinical trials from the literature, providing a clear and insightful analysis of the potential therapeutic applications. The figures and table in the paper are clear, well-designed, and informative for the reader's understanding.
The discussion is well-balanced, highlighting both the potential and challenges in targeting the Cyclin-CDK pathway. It also integrates molecular information and insights into potential therapeutic strategies, making it a valuable contribution in melanoma research.
Overall, this manuscript represents a good contribution to the field of melanoma studies and is therefore worth the publication.
Author Response
Thank you for the opportunity to review this very interesting paper. The paper offers a well-written and comprehensive review of the current knowledge regarding the Cyclin-D-CDK4/6-RB1 signaling pathway in melanoma. This is a critical research area given the high prevalence of alterations in this pathway and potentially an interesting therapeutical target. The authors have synthesized the most relevant studies and clinical trials from literature, providing a clear and insightful analysis of the potential therapeutic applications. The figures and table in the paper are clear, well-designed, and informative for the reader's understanding.
The discussion is well-balanced, highlighting both the potential and challenges in targeting the Cyclin-CDK pathway. It also integrates molecular information and insights into potential therapeutic strategies, making it a valuable contribution in melanoma research.
Overall, this manuscript represents a good contribution to the field of melanoma studies and is therefore worth the publication.
Thank you for your thoughtful and encouraging review. We truly appreciate your positive feedback on our manuscript, particularly regarding the clarity of our discussion, figures, and tables. Your recognition of the importance of the Cyclin-D-CDK4/6-RB1 pathway in melanoma and its therapeutic potential reinforces our motivation to contribute to this critical area of research. We greatly value your insights and support.
Round 2
Reviewer 1 Report
Comments and Suggestions for Authors
The authors have adequately addressed the first-round comments and improved the manuscript. Overall, it is comprehensive and a good candidate for publication in this journal.
Author Response
We appreciate your recognition of our efforts in addressing the first-round comments and improving the manuscript. We are glad to hear that you consider it a comprehensive manuscript and a good candidate for publication. We greatly value your insights and support.